# Biomonitoring of Atmospheric PAHs and PMs Using *Xanthoria parietina* and *Cupressus sempervirens* in Bouira (Algeria)

**Fatima Benaissa [1,2,3,*], Nassima Bourfis [4,5], Fatiha Ferhoum [4,5] and Isabella Annesi-Measano [1]**

1 Institute Desbrest of Epidemiology and Public Health (IDESP), University of Montpellier and INSERM, Montpellier Department of Pneumology, Allergology and Thoracic Oncology, Monpelier University Hospital, 34090 Montpellier, France; isabella.annesi-maesano@inserm.fr
2 Laboratory Biomathematics, Biophysics, Biochemistry and Scientometry L3BS, University of Bejaia, Targa Ouzemour, Bejaia 06000, Algeria
3 Common Core Department of Natural and Life Sciences, Targa Ouzemour, University of Bejaia, Bejaia 06000, Algeria
4 Department of Agricultural Sciences, Bouira University, Bouira 10000, Algeria; n.bourfis@univ-bouira.dz (N.B.); f.ferhoum@univ-bouira.dz (F.F.)
5 Food Technology Research Laboratory, Boumerdes University, Boumerdes 35000, Algeria
* Correspondence: fatima.benaissa@univ-bejaia.dz

**Abstract:** Air pollution constitutes a major environmental risk factor for living beings. Protection against such risk needs air pollution monitoring and control. Air pollution monitoring can be obtained in several ways. Amongst them, passive methods assessing cumulative exposure are of particular interest. A passive approach consisting of ambient concentrations biomonitoring of polycyclic aromatic hydrocarbons (PAHs) using lichens and plants was used for assessment of ambient air pollution exposure in the industrial region of Oued El Berdi in Bouira (Algeria). Seven stations were chosen to take samples of lichen thalli of *Xanthoria parietina* and conifer scales and barks of *Cupressus sempervirens* in April 2018. The physiological parameters of the chlorophyll and the proline content were measured, and the atmospheric PAHs and particulate matter (PM) concentrations were quantified. The results show a spatial variation between the different stations and directions. The PAH concentrations accumulated in lichen range from $35 \pm 3$ ng/g dw to $2222 \pm 376$ ng/g dw and show significant differences ($p = 0.017$). These concentrations are higher than those found in conifer scales ($18.8 \pm 7$ dw to $1183.5 \pm 876$ ng/g dw) and that found in conifer barks ($7 \pm 3$ dw to $515.3 \pm 19$ ng/g dw). Significant difference between the reference stations of Tikjda and Errich and the five industrial stations of Oued El Berdi were also observed. Physiological parameters (chlorophyll a, chlorophyll b, chlorophyll ab) and proline and air pollutants accumulated (PAHs and PM) were associated. Biomonitoring allowed to show that the industrial area of Oued El Berdi was impacted by PAHs and PM, which are generated mainly by factories located there.

**Keywords:** bioaccumulators; biomonitoring; polycyclic aromatic hydrocarbons (PAHs); PM; *Xanthoria parietina*; lichens; *Cupressus sempervirens*

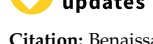



## 1. Introduction

Levels of air pollution are increasing worldwide due to the growth of man-made activities from industrial technology, energy production, domestic combustion, and intense road traffic. Air pollution affects the environment as well as human, animal, and vegetal health. Due to their immobility, vegetation is exposed to air pollution more than humans and animals [1].

Via their leaf system (from both cuticle and stomata) and their barks, plants directly accumulate air pollutants. Air pollutants accumulation can generate effects on leaf morphology, on the pigments content, on soluble sugar, and on proline [2] and chlorophyll [3].

Ascorbic acid, proline, and soluble sugar are no doubt important biochemical parameters that are not only needed by plants for growth, but the plants also need them to thrive when exposed to pollution stress [4].

Regarding morphological adverse effects, various studies found that the leaf and stomata decrease in size and increase in density in the plants from highly polluted sites [5–11]. Other studies have found a reduction in chlorophyll content in plants growing on urban roadsides [2,12,13] and in industrial areas [14–16]. It has also been observed that plants in the polluted regions have lower concentrations of starch and total and soluble sugars than plants in the unpolluted regions [17].

Among organic particulate pollutants, polycyclic aromatic hydrocarbons (PAHs) are a class of highly persistent environmental chemicals that occur naturally in coal, crude oil, and gasoline. They also are produced when coal, oil, gas, wood, garbage, and tobacco are burned. PAHs are very toxic pollutants that affect humans, animals, and plants due to several patterns of penetration. PAHs generated from the various sources can bind to or form small particles in the air. PAHs presence in plant tissues is due to the deposition on the leaves of the plants of airborne particulate matter (PM) that contains PAHs [18]. In plants, PAHs impinge growth, induce necrosis, abscission, epinasty, and chlorosis, and inhibit the photosynthetic processes as shown by many studies [19–23].

The monitoring of PAHs is becoming a necessity in environmental studies due to their mutagenic and carcinogenic properties that can seriously threaten human health [24]. In this context, biomonitoring, as a convenient alternative to the more expensive traditional sample collection, is becoming more and more popular, especially in remote areas where they may provide a measure of integrated cumulative exposure over an extended period of time [25].

The methods using plants for biomonitoring of air quality may turn out to be successful, as they are economical environmental tools and can supplement the classical physico-chemical methods [26].

So far, assessment of the levels of PAHs through plant biomonitoring have been limited to developed countries [27–31] and some other developing countries such as China [32]. No investigations were conducted in Africa. Lichens constitute another biomonitoring approach. This method was used to detect road traffic pollution [33] as shown by studies conducted in several countries, including in Algeria [34]. In other studies, this method was used for the biomonitoring of air pollution derived from industry, landfills, incinerators, etc. [35–37].

In this context, the present study aims to conduct a passive approach of PAHs assessment through bioaccumulation on a plant species, *Cupressus sempervirens* (a bioaccumulative, perennial, local species resistant to adverse climatic conditions) and one lichen species, *Xanthoria parietina* (which reacts to smaller doses of pollutants and thus makes it possible to characterize the state of an ecosystem in a practical and safe way) in the industrial zone of Oued El Berdi, Bouira in Algeria where sustainable development requires a consistent supply of fuel. These two species were selected based on characteristics of bioaccumulation defined by the literature [33,38]. They are easily identifiable and ubiquitous well-known bioaccumulators of air pollutants.

The aim of the present study is to assess air pollution in the industrial area of Oued El Berdi, Algeria. It consists of comparing PAH concentrations accumulated in *Xanthoria parietina* to those accumulated in *Cupressus sempervirens* (scales and barks) and to that accumulated in soil.

This is also consists of comparing PM concentrations between the different compartments (*Xanthoria parietina* and *Cupressus sempervirens* (scales and barks separately)) except the soil.

Another aim is to measure the effect of these pollutants on vegetation in dosing chlorophyll and proline content in our lichen and our conifer scales.

## 2. Materials and Methods

The physiological parameters of chlorophyll and proline were also studied. Thus, for proline concentrations, all lichen samples were compared with conifer scales and conifer barks. Regarding chlorophyll, concentrations in lichen were compared with conifer scales concentrations.

The flowchart in Figure 1 presents the research methodology.

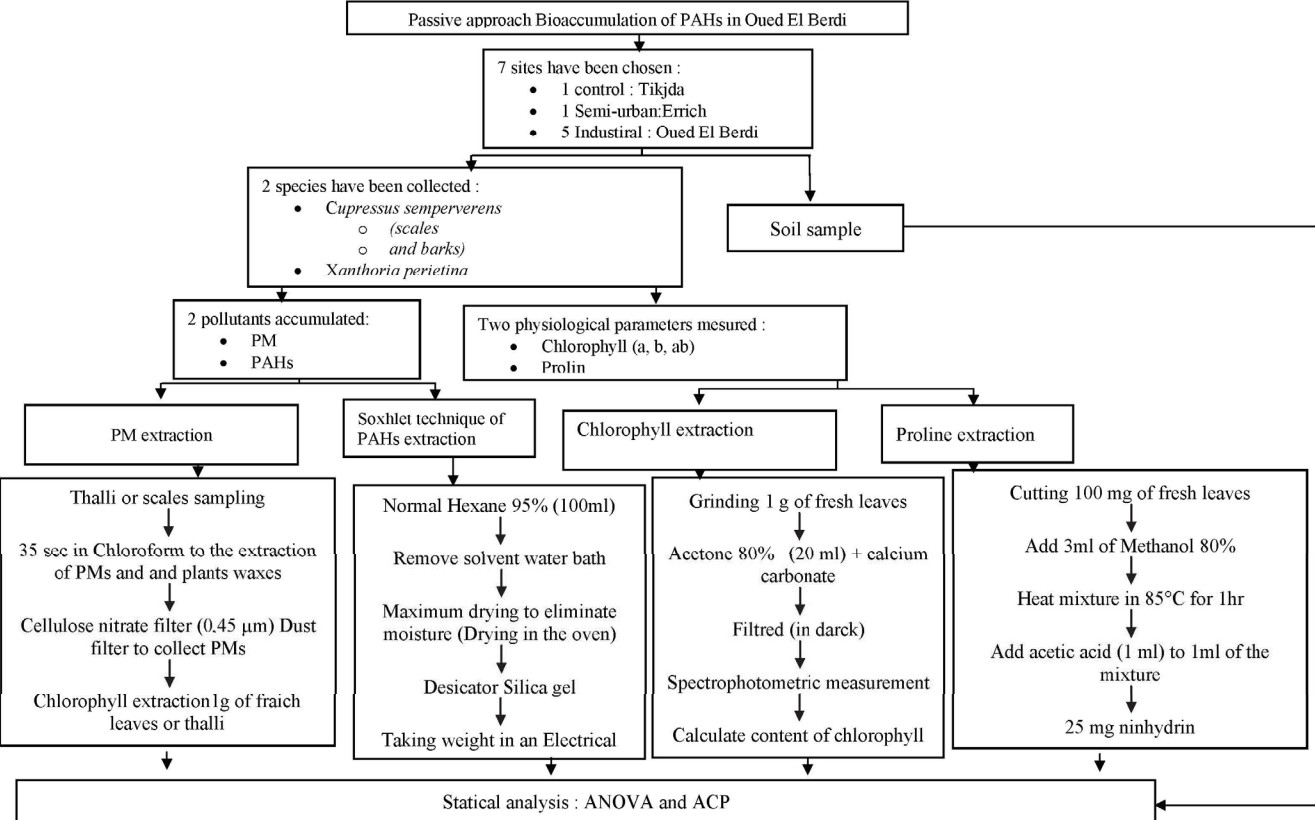

**Figure 1.** Flow chart of research methodology.

### 2.1. Description of the Area

Oued El Berdi is an industrial area of about 11,500 ha, located at 10 km away from Bouira department and at an altitude between 600 and 800 m in Algeria. It covers two main geomorphological groups (the high hills area in the south and the plateau area in the north). It has a hot and dry, temperate, Mediterranean climate, with average annual temperatures of between 5 and 30 °C and an abundant rainfall in winter (average annual precipitation of 564 mm). Using observational data over several decades, it is shown that the prevailing winds in the studied region have north-western and north-eastern direction in winter and autumn and have south-western directions in summer. For the purpose of the present study, seven sites were chosen to collect the samples from April to June 2018.

The study was mainly developed in the industrialized region of Oued El Berdi. So, five sites SOB1 (Oued El Berdi 1…5) were chosen in there to take samples according to the presence of the studied species (*Xanthoria parietina* and *Cupressus sempervirens*). The description of these sites is detailed in Table 1.

For the caparison, samples of studied species were taken from the control site of Tikejda (TS) and the urban site of Errich (Figure 2).

**Table 1.** Localization and characterization of sampling sites.

| Sites | Latitude | Longitude | Altitude (m) | Type | Vegetation Cover |
|---|---|---|---|---|---|
| (SOB1) | 361,718.23 | 35,310.23 | 650 | Industrial | Very rare |
| (SOB2) | 361,738.91 | 35,337.09 | 625 | Industrial | Rare |
| (SOB3) | 361,729.58 | 35,415.12 | 600 | Urban | More or less dense |
| (SOB4) | 361,655.57 | 3546.59 | 630 | Industrial | Low |
| (SOB5) | 361,618.74 | 35,332.84 | 645 | Industrial | Rare |
| Errich (ES) | 362,437.25 | 35,240.95 | 610 | Semi-urban | Very dense |
| Tikjda (TS) | 362,650.95 | 40,733.64 | 1400 | Forest | Very dense |

SOB1: Site Oued El Berdi 1…5, ES: Errich site, TS: Tikjda site.

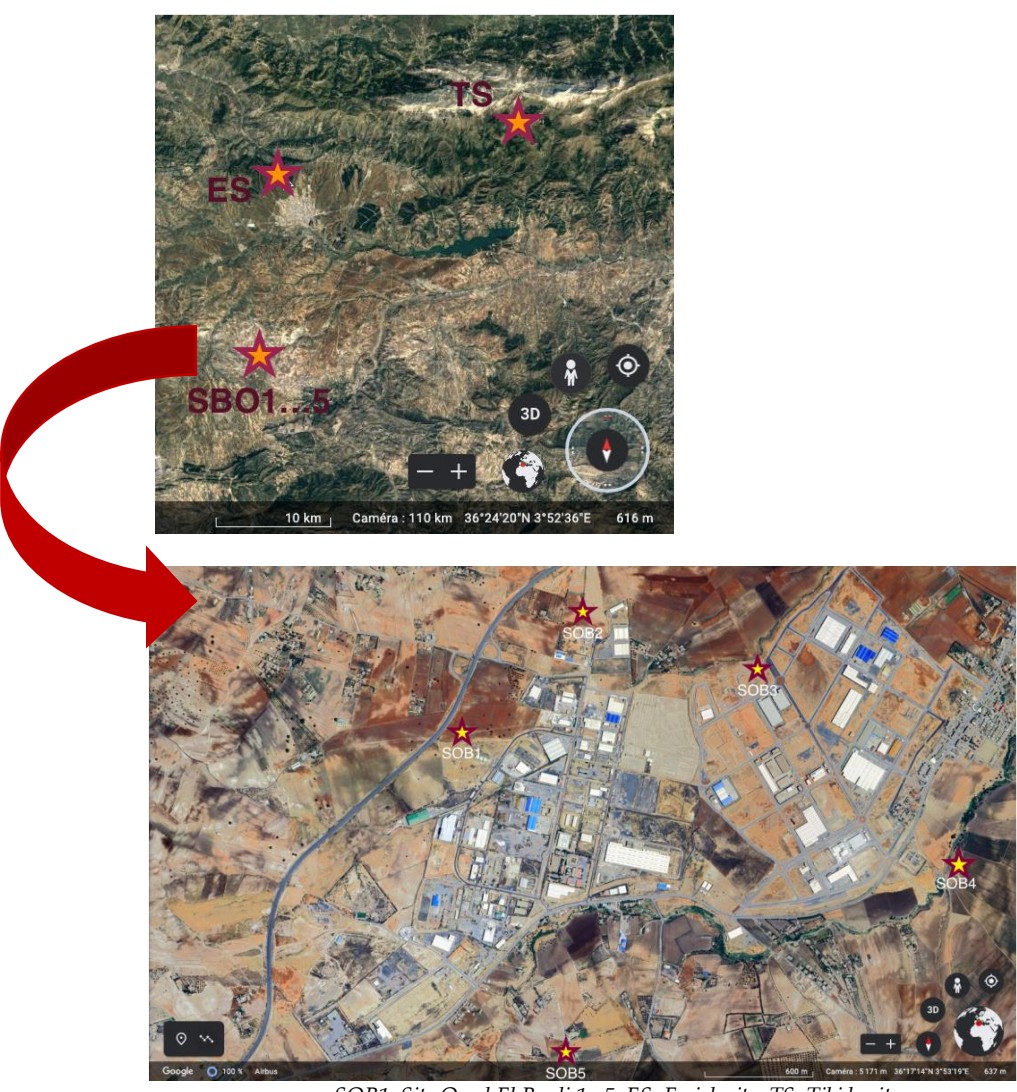

*SOB1: Site Oued El Berdi 1...5, ES: Errich site, TS: Tikjda site*

**Figure 2.** Location of the monitoring sites.

## 2.2. Sampling Techniques

Forty-one samples of *Xanthoria parietina* and *Cupressus sempervirens* were used for the assessment.

### 2.2.1. *Xanthoria parietina* Samples

In each station, 6 to 12 lichen thalli were taken from 3 to 6 *Olea europaea* trees. The collection was made from trunks at the four cardinal points of each tree and at a height of

1.5–2 m using a ceramic knife. The collected samples were packed in paper bags and transported in a cooler to protect them from sunlight and to keep them fresh. All samples were taken after a period of at least 20 days without rain, suggesting that PAHs are retained within the lichen thalli and are not rinsed off.

### 2.2.2. *Cupressus sempervirens* Samples

From the same lichen sampling sites, 3 to 5 twigs (10 to 20 scales) of *Cupressus sempervirens* were collected using a pruner, from the terminal part of branches. To avoid conifer trunk damage, only a few of cm$^2$ of each of the four cardinal points of the trunk were taken.

The bark samples were taken at chest height along the four cardinal points. They only concern a few cm$^2$ on each of the four points of the trunk, so as not to damage it too much. The samples collected were put in paper bags and transported in a cooler [5].

### 2.2.3. Soil Samples

In order to increase the performance of the diagnosis, the soil analysis was used simultaneously. In parallel to lichen and conifer sampling sites, soil samples were collected from the upper 5 cm of soil and putt in polyethylene bags. Once in the laboratory, soil samples were sieved through a 2 mm mesh screen to remove large solid fragments and plant debris. To avoid the volatility of PAHs, soil samples were dried as carefully as possible in an oven at a temperature below 40 °C for 48 h. Then, the sieved soil samples were transferred to glass bottles in order to prevent adsorption by plastic, protected from sunlight and stored at 4 °C. This is in accordance with NF ISO 10381-5 standard, described by [18].

### 2.3. Air Pollutants Extraction

Dust deposited on plant leaves reduces chlorophyll contents in there. The chemical constituents of PM$_{2.5}$ associated with polycyclic aromatic hydrocarbons (PAHs) may contribute to reductions in chlorophyll (Ch).

To measure the quantity of atmospheric pollutants, namely PM and PAHs, captured by the vegetation of the studied sites, and to see how this accumulation has an effect on the physiological activity of the plants, we used different extraction techniques and dosage.

The method of PMs extraction was that described by Garrec [5]. So, the extraction was made using 10 g of fresh material and chloroform solvent. To dissolve the wax, samples were soaked for 35 s in chloroform and then filtered in a cellulose nitrate filter (diameter 0.45 μm). The filters were dried in 40 °C for about 24 h and weighted using a precision balance. Before determining their concentration, PAHs were extracted from each sample. The extraction was performed using the Soxhlet extractor, model Behrotest. Thus, a mixture of 100 mL of acetone and 100 mL of hexane was added to 10 g of sample and was placed in a cellulose cartridge previously cleaned with acetone. After 3 h of extraction, the Soxhlet crucibles were placed in a water bath at 90 °C to evaporate the solvents. The sample was then put in an oven for the drying of the hydrocarbons. After removal of moisture, the extracts were deposed in a desiccator silica and were weighed by a precision balance.

### 2.4. Chlorophyll and Proline Extraction

The extraction of the chlorophyll was carried out by crushing 1 g of fresh leaf tissue and suspended in test tubes containing calcium carbonate and acetone (20 mL at 80%). The resulting solution was filtered in the dark to avoid the oxidation of the chlorophyll. The absorbance was read in a spectrophotometer (EZ Swing 3k) at 645 and 663 nm. Chlorophyll a (Chl a), chlorophyll b (Chl b), and total chlorophyll (Chl a + b) were calculated by using formulates taken from the literature and given below:

Chl a: 12.7 (DO663) − 2.69 (DO645).
Chl b: 22.9 (DO645) − 4.86 (DO663).
Chl a + b: 8.02 (DO645) + 20.20 (DO663).

To determine proline content, according to the method described by Monneuveux and Nemmar (1986) [39], 100 mg of cut material were placed in test tubes. After that, 3 mL of 80% methanol was added in the tubes, and the mixture was heated in a water bath at 85 °C for 1 h.

The mixture of 1 mL solution, 1 mL of acetic acid, and 1 mL was added to another mixture of 120 mL of distilled water, 300 mL of acetic acid, 80 mL of orthophosphoric acid, and 25 mg of ninhydrin.

Then, the resulting mixture was brought to a boil for 30 min when red color was obtained. The solution was cooled again and then was stirred in the presence of 5 mL of toluene. The upper phase obtained by stirring was sucked and dehydrated in the presence of $Na_2SO_4$. The samples were then assayed using a spectrophotometer (EZ Swing 3k) at a wavelength of 528 nm.

The considered variables include physiological parameters ($Ch_a$, $Ch_b$, $Ch_{ab}$) and proline and air pollutants accumulated (PAHs and PM).

### 2.5. Statistical Analysis

Descriptive statistics (median, mean, standard deviation, minimum and maximum) were used to characterize PAH and PM concentrations determined in lichens, conifer scales, and barks.

A two-way ANOVA test was used to compare PM and PAH concentrations of different sites, different directions, and the interaction between sites and directions. This was followed by the post hoc Tukey's test to find out which specific sites' means are different.

A principal component analysis (PCA) was performed in order to evaluate the pollutants accumulated in different indicators. This analysis was undertaken using physiological parameters and pollutants.

All statistical tests were performed using R statistical software version 3.3 (R Core Team, Vienna, Austria, 2016). All *p*-values were two-tailed. $p < 0.05$ was considered statistically significant.

## 3. Results

### 3.1. Estimation of Accumulated PAHs

The statistical summary of PAHs and PMs concentrations in different conifer parts (barks and scales), lichens, and soil is presented in Table 2.

**Table 2.** Statistical PAHs summary.

| PAHs (ng/g dw) | *Xanthoria parietina* | | | | | Conifer Scales | | | | | Conifer Barks | | | | | Soil | | | | |
|---|---|---|---|---|---|---|---|---|---|---|---|---|---|---|---|---|---|---|---|---|
| | Min | Me | M | SD | Max | Min | Me | M | SD | Max | Min | Me | M | SD | Max | Min | Me | M | SD | Max |
| TS | 21.5 | 32.1 | 35.6 | 3 | 58.5 | 9.9 | 20.2 | 18.8 | 7.0 | 30.3 | 3 | 9 | 7 | 3 | 13 | 8.7 | 9.6 | 9.9 | 1.1 | 10.6 |
| ES | 102 | 200 | 229.9 | 33.2 | 267 | 12.2 | 17.3 | 21.6 | 13.0 | 44.2 | 0 | 7 | 8 | 11 | 28.4 | 70 | 73.9 | 75.3 | 0.2 | 76.5 |
| OBS1 | 429.6 | 993.1 | 1091.1 | 224 | 1988 | 183.2 | 416.4 | 387.2 | 118.0 | 690.8 | 88.9 | 290.3 | 302.6 | 130 | 530 | 148 | 150.6 | 159.7 | 4.8 | 162 |
| OBS2 | 443.1 | 545 | 588.8 | 192 | 765.4 | 99.6 | 296.3 | 276.1 | 95.0 | 290.5 | 50.4 | 157.3 | 163.3 | 79 | 186.1 | 59.5 | 60.1 | 59.4 | 2.4 | 64.4 |
| OBS3 | 1400.1 | 2154.1 | 1901.6 | 590 | 2668 | 331.3 | 544.5 | 538.2 | 257.0 | 766.4 | 256.1 | 320.1 | 340.3 | 111 | 608.4 | 87.1 | 100.3 | 97.6 | 29 | 108 |
| OBS4 | 280 | 703.3 | 744.4 | 461 | 921.3 | 497.2 | 577.1 | 537.9 | 160.0 | 642.1 | 14.4 | 19.5 | 21.4 | 41 | 50.6 | 73.6 | 75.5 | 76.2 | 2.7 | 80.1 |
| OBS5 | 1967.1 | 2099.9 | 2222.2 | 376 | 2695.4 | 2922 | 2176.4 | 1183.5 | 876 | 9012.5 | 498.9 | 500.2 | 515.3 | 19 | 574 | 119 | 129.6 | 138.2 | 37 | 140 |

PAH: polycyclic aromatic hydrocarbons, TS: Tikjda Site, ES: Errich Site, OBS1: Oued El Berdi Site 1...5. M: mean, Me: median, SD: Standard deviation.

### 3.1.1. PAHs in *Xanthoria parietina*

The highest value of PAHs accumulated in *Xanthoria parietina* (2222 ± 376 ng/g dw) was recorded at SOB5. It was 66 times higher than the one found in the control site of Tikjda, which is 35 ± 3 ng/g dw (Figure 3). Two-way ANOVA test (Table 3) revealed a highly significant effect of direction on the accumulation ($p < 0.001$) and significant variations of effect between sites ($p = 0.017$). Then, ANOVA was followed by the post hoc Tukey test to

explore differences between site means while controlling for the experiment-wise error rate. It was the control site of Tikjda that showed high differences with all the other sites.

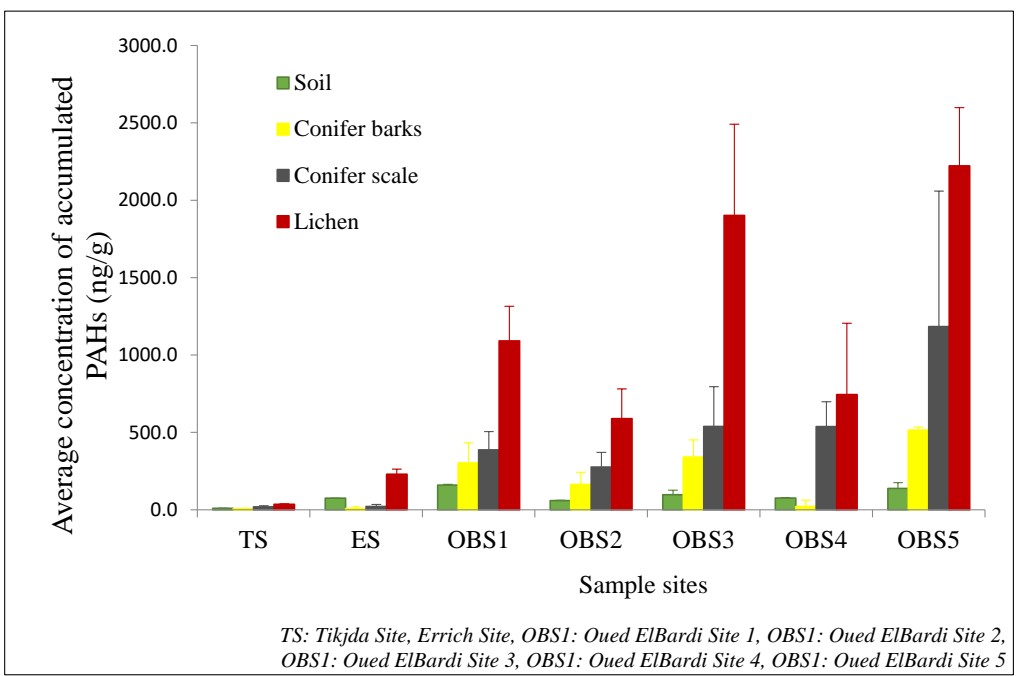

**Figure 3.** Mean and standard deviation of the content (in ng/g) PAHs accumulated by the lichen (*Xanthoria parietina*), conifer scale, conifer bark and the soil in different sample sites.

**Table 3.** Two-way ANOVA.

|  | PAHs | | PMs | | Cha | | Chb | | Cha + b | | Proline | |
|---|---|---|---|---|---|---|---|---|---|---|---|---|
|  | **F** | *p* | **F** | *p* | **F** | *p* | **F** | *p* | **F** | *p* | **F** | *p* |
| **D** | 10.24 | *** | 4.91 | 0.002 | 17.17 | *** | 33.78 | *** | 25.84 | *** | 18.68 | *** |
| **Sites** | 4.1 | *** | 3.61 | *** | 22.28 | *** | 35.34 | *** | 23.81 | *** | 50.94 | *** |
| **Inter** | 3.02 | *** | 1.83 | 0.005 | 7.60 | *** | 11.90 | *** | 8.35 | *** | 5.58 | *** |

*** $p < 0.001$, D: Direction.

### 3.1.2. PAHs in *Cupressus sempervirens*

As shown in (Table 2), concentrations of PAHs in *Cupressus sempervirens* scales ranged from $18 \pm 7$ to $1183 \pm 876$ ng/g dw. The site of OBS5 presented the highest concentration. On comparing with the control site of Tikjda, the average PAHs concentration found in the OBS5 site was 65 times higher than that was found in Tikjda ($18.8 \pm 7$ ng/g dw).

The lowest average concentration calculated in the industrial site was also matched higher than the control site; this was found in the OBS1 site ($276 \pm 95$ ng/g dw).

When taking the two parameters of sites and directions in the two-way ANOVA test, it was given that the west direction of the OBS5 was the more exposed of the two pollutants.

### 3.1.3. PAHs in the Soil

Once again, the highest concentration of PAHs in the soil was found in the industrial site of OBS1, with $159.7 \pm 4.8$ ng/g dw followed by OBS2 with $59.4 \pm 2.4$ (ng/g dw), while the lowest with $9.9 \pm 1.1$ was found in the TS site.

In this case, the ANOVA test gave a highly significant difference between sites with an F = ($p = 0.004$). Tukey's test gave $p < 0.001$ for each site combinations.

### 3.2. Estimation of Dust Deposited on Lichen and Conifer

3.2.1. PM Concentration in *Xanthoria parietina* Thalli

Apart from the two sites of OBS1 and OBS2, all other sample sites presented low PM deposed and accumulated concentrations in lichen thalli. Indeed, as shown in Table 4, the highest average value of 5763.3 ± 678.3 µg/kg, which was calculated for the site of OBS1, is 16 times higher than that calculated for the site of OBS4 (366.7 ± 109.8 µg/kg).

**Table 4.** Statistical PMs summary.

| PMs (µg/g/dw) | *Xanthoria parietina* | | | | | Conifer Scales | | | | | Conifer Barks | | | | |
|---|---|---|---|---|---|---|---|---|---|---|---|---|---|---|---|
| | Min | Me | M | SD | Max | Min | Me | M | SD | Max | Min | Me | M | SD | Max |
| TS | 1879.3 | 2115.2 | 2213.3 | 113.1 | 2680.2 | 36.6 | 50.1 | 54.1 | 45.6 | 87.6 | 55.8 | 68.3 | 72.9 | 13.4 | 103.3 |
| ES | 3110.5 | 3765.1 | 3713.3 | 21.9 | 3801.3 | 95.4 | 104.3 | 162.6 | 54.9 | 210.6 | 73.5 | 128.4 | 125.2 | 63.6 | 201.9 |
| OBS1 | 3090.4 | 5709.6 | 5763.3 | 678.3 | 6984.5 | 1019 | 1506.4 | 1474.2 | 795 | 1659.4 | 3020.1 | 4321.7 | 4631.6 | 428.9 | 4861.1 |
| OBS2 | 1983.9 | 3354.3 | 3468.9 | 850.0 | 5769.6 | 1879.3 | 2343.6 | 2529.1 | 780 | 3665.5 | 5331.7 | 5692.1 | 5766.6 | 434.7 | 6210.1 |
| OBS3 | 577.4 | 654.4 | 700.0 | 466.6 | 899.6 | 398.4 | 488.5 | 511.2 | 188.1 | 843.1 | 222.8 | 267.5 | 273.7 | 63 | 376.2 |
| OBS4 | 102.5 | 348.7 | 366.7 | 109.8 | 521.2 | 264.5 | 255.1 | 295.5 | 87.6 | 411.4 | 98.9 | 120.1 | 121 | 18.1 | 157.7 |
| OBS5 | 820.4 | 1032.3 | 983.3 | 123.1 | 1178.6 | 118.4 | 181 | 201.6 | 92.8 | 465 | 132.2 | 147 | 143.3 | 23.7 | 183.3 |

PM: Particulate Matter, TS: Tikjda Site, ES: Errich Site, OBS1: Oued El Berdi Site 1...5. M: mean, Me: median, SD: Standard deviation.

This difference was confirmed by two-way ANOVA test in Table 3. It showed a highly significant difference between sites (F = 4.1 and $p < 0.001$) and a significant difference between directions (F = 10.24, $p = 0.036$).

3.2.2. PM Concentration in *Cupressus sempervirens*

In the case of PM extracted from conifer scales and in concordance with the lichen results, in the two industrial sites, it was the south direction of the site OBS2 that recorded the highest value of 2529.1 ± 780 µg/kg; it was 277 times higher than that found (54.1 ± 45.6 µg/kg).

The average concentration of PM in conifer barks was found in order, SOB2 > SOB1 > SOB3 > SOB4 > SOB5 (Figure 4).

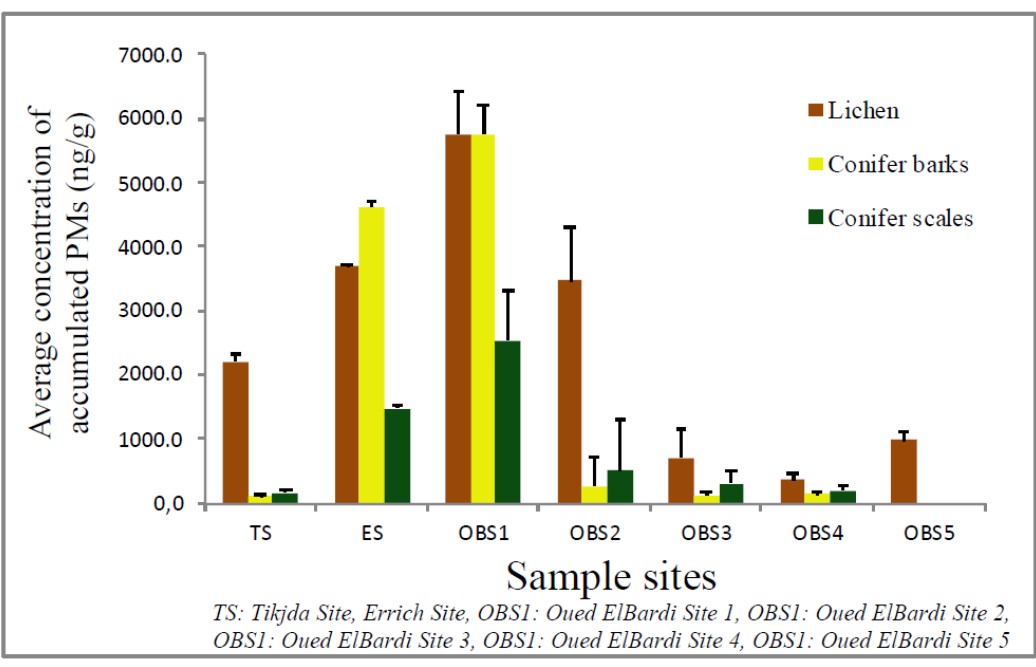

TS: Tikjda Site, Errich Site, OBS1: Oued ElBardi Site 1, OBS1: Oued ElBardi Site 2, OBS1: Oued ElBardi Site 3, OBS1: Oued ElBardi Site 4, OBS1: Oued ElBardi Site 5

**Figure 4.** Mean and standard deviation of the PMs content (in ng/g) in lichen. (*Xanthoria parietina*), in conifer (*Cupressus sempervirens*) scale and barks in different sample sites.

### 3.3. Physiological Parameters

3.3.1. Chlorophyll Variations

In the case of *Xanthoria parietina,* the results of physiological parameters including chlorophyll (a,b and a + b) range successively from (165.5 ± 74.2, 259.8 ± 95.4, 425.3 ± 109.2 μg/kg) in OBS2 site to (429.8 ± 9.8, 774.5 ± 11.8, 1204.4 ± 19.8 μg/kg) in the control site of TS.

In parallel, chlorophyll (a,b and a + b) concentrations calculated from conifer scales show successively low values of 323.2 ± 75.3, 571.5 ± 133.2, 894.8 ± 207.2 and high values of 434.1 ± 20.1, 694.6 ± 30.2, 1128.7 ± 35.5 μg/kg.

Similarly to the variations found in the case of the lichen, the lowest values for the conifer scales were found in the site OBS2, and the highest values were found in the control site of TS (Figure 5).

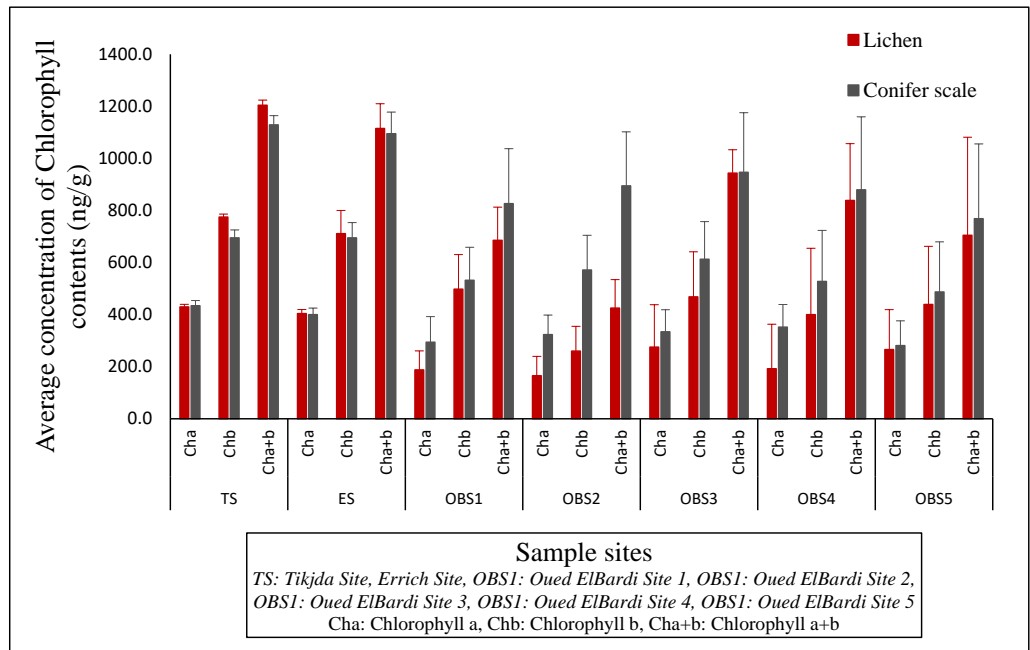

**Figure 5.** Mean and standard deviation chlorophyll content (in ng/g) in lichen (*Xanthoria parietina*) and in conifer (*Cupressus sempervirens*) scale in different sample sites.

3.3.2. Proline

However, the physiological parameter of proline shows an opposite variation to that of chlorophyll. So, the highest average value (529.1 ± 34.4 μg/kg) was recorded at site OBS2 followed by OBS1 with an average value of (525.0 ± 48 μg/kg), then OBS5 (457.9 ± 110.6 μg/kg) and OBS4 (488.1 ± 44.6 μg/kg) (Figure 6).

The lowest average value of proline calculated in the lichen (184.9 ± 0.3 μg/kg) and that found in TS is close to that (185.2 ± 0.5 μg/kg) found in the second control site of ES in the case of conifer scales.

### 3.4. Comparison between Xanthoria parietina, Cupressus sempervirens, and Soil

The two axes of principal component analysis explained 80.02% of the information contained in our input table. The first axis formed by pollution parameters explains 63.46%, and the second axis formed by physiological parameters explains 16.56%.

The three vectors of chlorophyll a (Chl a), chlorophyll b (Chl b), and total chlorophyll (Chl a + b) are close. So, these variables are positively correlated with one other and inversely correlated with the variable of proline (Table 5).

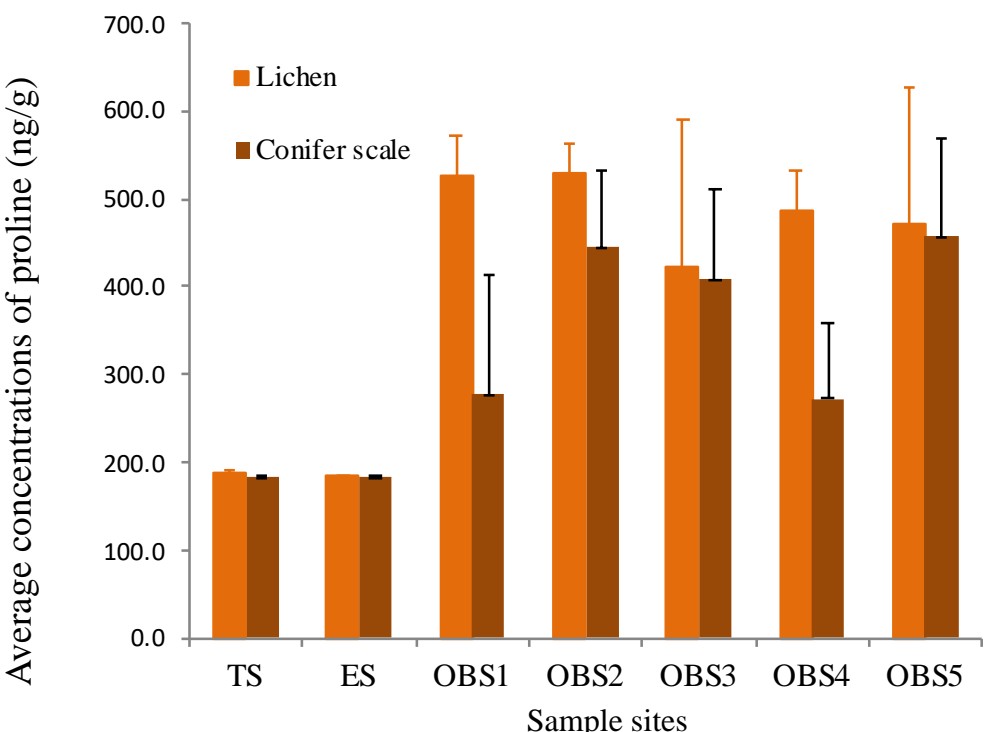

*TS: Tikjda Site, Errich Site, OBS1: Oued ElBardi Site 1, OBS1: Oued ElBardi Site 2,
OBS1: Oued ElBardi Site 3, OBS1: Oued ElBardi Site 4, OBS1: Oued ElBardi Site 5*

**Figure 6.** Mean and standard deviation the proline content (in ng/g) in lichen (*Xanthoria parietina*) and conifer scale in different sample sites.

**Table 5.** Correlation matrix of PCA's variables.

| Variable | Ch.a | Ch.a + b | Ch.a | PAH | PM | Proline |
|---|---|---|---|---|---|---|
| **Ch.a** | 1 | | | | | |
| **Ch.a + b** | 0.92 | 1 | | | | |
| **Ch. B** | 0.84 | 0.95 | 1 | | | |
| **PAH** | −0.60 | −0.64 | −0.63 | 1 | | |
| **PM** | −0.47 | −0.47 | −0.43 | 0.3 | 1 | |
| **Proline** | −0.79 | −0.80 * | −0.79 | 0.65 * | 0.4 | 1 |

* $p < 0.05$. PCA shows proline is negatively correlated with PAHs concentrations (r = 0.65, $p < 0.001$) and with PM concentrations (r = 0.4, $p < 0.001$).

## 4. Discussion and Conclusions

In this paper, we presented a passive approach of air pollution biomonitoring in an industrial area of Oued El Berdi in Bouira, Algeria. Findings support previous results, having shown the usefulness of lichens and conifers biomonitoring to estimate the local concentrations of PAHs and PMs [11,29].

The highest PAH levels are recorded at the west direction of the site OBS5 due to its location near the Africa Bitume industrial unit. However, the results in the site of Errich (ES) appear to conflict with its urban nature of high pollution. This could be due to the presence of dense vegetation in this site.

The assessment of PAHs quantities accumulated by different plant parts presented remarkable significant differences.

The results recorded in industrial area of Oued El Berdi and concerning the high concentrations of PAHs and PMs accumulated in *Xanthoria parietina*, *Cupressus sempervirens*, and soil reveals that this zone is more than polluted. Similarly to the pollutants concen-

trations, physiological parameters of proline content that show the highest values and chlorophyll content that show the lowest values reveal the same hypothesis.

The high concentrations of PAHs and PMs accumulated in lichen and conifer suggest that they can be used as pertinent biomonitors of air pollution.

When comparing the results found in our study with others presented in Table 6, it is evident that our sites are polluted.

Regarding PAH concentrations, the mean values calculated for *Xanthoria parietina* ($35.6 \pm 3$–$2222 \pm 376$ µg/kg dw) are twice as high as those for conifer scales ($18.8 \pm 7$ to $1183.5 \pm 876$ ng/g dw) and four times higher than those for conifer barks $0.07 \pm 0.03$ and $515.3 \pm 19$ ng/g dw.

Studies from Spain [40] and Portugal [41] on lichens have reported successively lower values of total PAH concentrations (108–330 ng/g dw and 95.5–873.8 ng/g dw) than those found in our sites.

The PAH concentrations ($18.8 \pm 7$–$1183.5 \pm 876$ µg/kg dw) found in scales of the conifer used in this study, are within the same range of those found in other studies in which concentrations range from 83.0–466.8 µg/kg dw with a mean of ($198.9 \pm 184.2$ µg/kg dw) in Japan [42] and about $185.4 \pm 113.5$ µg/kg dw in Portugal [41].

PAH accumulated in conifer barks of the current study, which range from $0.07 \pm 0.03$ to $515.3 \pm 19$ µg/kg dw lower than those found in Italy, which range from 33 to 1015 µg/kg [43].

All of our soil samples can be classified as contaminated when PAH concentrations in there are between $9.9 \pm 1.1$ and $138.2 \pm 37$ µg/kg dw and so upper the standard value (200 µg/kg dw) given by Maliszewska-Kordybach [44].

PAHs presence in plant tissues is due to the deposition on the leaves of the plants of airborne particulate matter (PM) that contains PAHs [18]. This is confirmed with conifer's scales and barks.

In the case of lichen, the OBS 5 site, which had the highest PAH concentrations, had PM concentrations lower than those found in the two control sites.

It is known that PM contains PAHs but also known that these particles come in many shapes and can be made up of hundreds of different chemicals [45,46].

**Table 6.** Summary of the current study results compared to others in the literature.

| | Parameters | Current Study | Results from Literature |
|---|---|---|---|
| Lichen | PAHs (ng/g dw) | (35.6–2222) | Spain (108 and 330) [40], Portugal (95.5–873.8) [41] |
| | Chl (ng/g dw) Proline | (425.3–1204.4) (184–597) | Algeria (1870) [47], Algeria (50–380) [48] |
| Conifer | PAH (g/kg dw) | Scales (18.8 et 1183.5) Barks (07–515.3) | Japan (83.0–466.8) [42], Portugal ($185.4 \pm 113.5$) [41], Italy (33–1015) [43] |
| | Chl rate (ng/g dw) | Scales (894.8–1128.7) | India (2250–3640) [10] |
| Soil | PAHs (ng/g dw) | (9.9–138.2) | (0.8 and 30) [44]. |

Particulate matter (PM) includes bioaerosols (pollen, fungal spore, bacteria, viruses, etc.) and non-biological particles.

So, if in the control site of Tikjda, the PM concentrations were higher than in the OBS 5 site, it is because the PM found in lichen sampled in Tikjda do not have the same origin as that found in OBS5.

In the present work, *Xanthoria parietina* has been used as a biosensor. In addition, two physiological parameters (chlorophyll and proline contents) were also measured. Concentrations of these parameters were higher in samples from the air polluted sites compared to the control.

The total chlorophyll accumulated in *Xanthoria parietina* of the current study recorded lower results ($425.3 \pm 109.2$–$1204.4 \pm 19.8$ ng/g dw) than that (1870 ng/g dw) recorded

by Khelil et al. in their work [47]. This is because they used an active technique of biomonitoring, not a passive one like that used in the present study.

Giri et al. [3] recorded higher levels of total chlorophyll in the leaves of Azadirachta indica (2250 $\pm$ 460 and 3640 $\pm$ 650 ng/g dw) than in the conifer of our study (894.8 $\pm$ 207.2–1128.7 $\pm$ 35.5 ng/g dw).

For proline and for the same period of time, Juin Maizi et al. in their work demonstrated nearly the same levels (50–380 ng/g dw) [48] with those of our work (184 $\pm$ 0.3–597 $\pm$ 48 ng/g dw).

This is a preliminary study that was carried out on the effect of air pollution on vegetation in the industrial region of Oued El Berdi (Bouira) using an economical and environmental method of biomonitoring.

We were unable to explain the relationships of PM concentrations and PAH concentrations in different sites of our study enough due to the fact that we did not know exactly the source of each of these pollutants. This needs chemical characterization the pollutants.

Further studies are required for a better estimate of the air pollution risk in the industrial area of Oued El Berdi, and it is necessary to carry out other PAH biomonitoring studies in this region using other bioindicating plants.

**Author Contributions:** F.B.: Conceptualization; Data curation; Formal analysis; Funding acquisition; Investigation; Methodology; Project administration; Resources; Software; Visualization; Roles/Writing—original draft; Writing—review and editing. N.B.: Methodology; Resources. F.F.: Methodology; I.A.-M.: Supervision; Validation. All authors have read and agreed to the published version of the manuscript.

**Funding:** This research received no external funding.

**Institutional Review Board Statement:** Not applicable.

**Informed Consent Statement:** Not applicable.

**Acknowledgments:** The authors are indebted to Rezak Alkama and to Laboratory Biomathematics, Biophysics, Biochemistry, and Scientometry L3BS, University of Bejaia, Targa Ouzemour, Bejaia, Algeria.

**Conflicts of Interest:** The authors declare no conflict of interest.

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
