# Peer review of "Biomonitoring of Atmospheric PAHs and PMs Using Xanthoria parietina and Cupressus sempervirens in Bouira (Algeria)"

_sustainability, doi:10.3390/su152015174_

Round 1

Reviewer 1 Report

There are several significant faults with the presented results and discussion. In the introduction, authors assert:

PAHs generated from the various sources can bind to or form small particles in the air. PAHs presence in plant tissues is due to the deposition on the leaves of the plants of airborne particulate matter (PM) that contains PAHs [17].

However, the OBS 5 site had the highest PAH concentrations, while the PM concentrations were among the lowest in the group, calling into question the primary hypothesis regarding PAHs transport mechanisms and their relationship with PM. There are also inconsistencies in the statistical data reported, such as in table 2, where the minimum PAHs concentration in conifer scales at OBS 5 is higher than the maximum concentration reported - 9012 versus 2922.

The same applies to other hypotheses presented, such as chlorophyll concentrations. In their introduction, the authors state:  

Other studies have found a reduction in chlorophyll content in plants growing on urban roadside [10-12] and in industrial area [13-15].

In contrast , the results presented show that chlorophyll (a,b and a+b) range successively from (165.5 ± 74.2, 259.8 ± 95.4 , 425.3 ± 109.2 231 μg/kg) in the control site of TS and from (429.8 ± 9.8, 774.5 ± 11.8, 1204.4 ± 19.8 μg/kg) at OBS2 site, showing significantly higher concentrations in industrial site. And this results didn’t correspond with the graph presented given in figure 4.

This creates confusion and casts doubt on the entire presentation.

No major changes are required.

Author Response

There are several significant faults with the presented results and discussion. In the introduction, authors assert:

PAHs generated from the various sources can bind to or form small particles in the air. PAHs presence in plant tissues is due to the deposition on the leaves of the plants of airborne particulate matter (PM) that contains PAHs [17].

However, the OBS 5 site had the highest PAH concentrations, while the PM concentrations were among the lowest in the group, calling into question the primary hypothesis regarding PAHs transport mechanisms and their relationship with PM.

We agree with this comment since we have not discussed this point sufficiently. Therefore, in the reviewed version,

  • We firstly added a paragraph (in discussion part of the manuscript) in which we explained the result found.

“PAHs presence in plant tissues is due to the deposition on the leaves of the plants of airborne particulate matter (PM) that contains PAHs [18]. This is confirmed with conifer’s scales and barks.

In the case of lichen, OBS 5 site which had the highest PAH concentrations, had PM concentrations lower than those found in the two control sites.

It’s known that PM contains PAHs but also known that these particles come in many shapes and can be made up of hundreds of different chemicals [40,41].

Particulate matter (PM) includes bioaerosols (pollen, fungal spore, bacteria, viruses, etc.) and non-biological particles.

So if in the control site of Tikjda, the PM concentrations were higher than in the OBS 5 site, it’s that PM found in lichen sampled in Tikjda have not the same origin as that found in OBS5”

Our samples were taken on April (the flowering period where vegetation is dense).   

  • We secondly, add the following sentence, in conclusion part of the manuscript:

“We were unable to explain enough the relationships of PM concentrations and PAH concentrations in different sites of our study due to the fact that we did not know exactly the source of each of these pollutants. This needs the chemical characterization the pollutants”

[45] Ortega-Rosas, C.I., Meza-Figueroa, D., Vidal-Solano, J.R. et al. Association of airborne particulate matter with pollen, fungal spores, and allergic symptoms in an arid urbanized area. Environ Geochem Health 43, 1761–1782 (2021). https://doi.org/10.1007/s10653-020-00752-7

[46] Liu, X.; Yu, X.; Zhang, Z. PM2.5 Concentration Differences between Various Forest Types and Its Correlation with Forest Structure. Atmosphere 20156, 1801-1815. https://doi.org/10.3390/atmos6111801

There are also inconsistencies in the statistical data reported, such as in table 2, where the minimum PAHs concentration in conifer scales at OBS 5 is higher than the maximum concentration reported - 9012 versus 2922.

Thank you for pointing this out. We reversed the measurements when copying them from Excel to Table 2. So now, as you can see in revised version of the paper, the minimum PAHs concentration in conifer scales at OBS 5 is  2922 ng/g and maximum concentration is 9012 ng/g.

The same applies to other hypotheses presented, such as chlorophyll concentrations. In their introduction, the authors state:  

Other studies have found a reduction in chlorophyll content in plants growing on urban roadside [10-12] and in industrial area [13-15].

In contrast , the results presented show that chlorophyll (a,b and a+b) range successively from (165.5 ± 74.2, 259.8 ± 95.4 , 425.3 ± 109.2 231 μg/kg) in the control site of TS and from (429.8 ± 9.8, 774.5 ± 11.8, 1204.4 ± 19.8 μg/kg) at OBS2 site, showing significantly higher concentrations in industrial site. And this results didn’t correspond with the graph presented given in figure 4.

This creates confusion and casts doubt on the entire presentation.

Thank you for pointing this out. We reversed the measurements when writing. So now, as you can see in revised version of the paper and as shown in figure 04, the results of lichen show that chlorophyll (a,b and a+b) range successively from (429.8 ± 9.8, 774.5 ± 11.8, 1204.4 ± 19.8 μg/kg) in the control site of TS and from (165.5 ± 74.2, 259.8 ± 95.4 , 425.3 ± 109.2 231 μg/kg) at OBS2 site.

Reviewer 2 Report

Dear authors,

you submitted the contribution to the journal "Sustainability" but apsects of sustainability are not part of the study. Meanwhile, you show that pollution near to industrial sites is higher than at reference sites. Really, that is not a news. The methods that you use also are well known. There is not an innovative approach. "Sustainability" also means to develop solutions for environmental problems. I miss it here.

Best regards

Your reviewer

Author Response

Response to reviewer 2 Comments

Thank you very much for taking the time to review this manuscript. Please find the detailed responses below and the corresponding revisions.

you submitted the contribution to the journal "Sustainability" but apsects of sustainability are not part of the study. Meanwhile, you show that pollution near to industrial sites is higher than at reference sites. Really, that is not a news. The methods that you use also are well known. There is not an innovative approach. "Sustainability" also means to develop solutions for environmental problems. I miss it here.

It is known that in Algeria as in other developing countries, there is a need for an effort to get a sustainable development strategy to improve air quality.

Thank you for pointing this out. We accordingly added a sentence “where sustainable development requires a consistent supply of fuel” to tell why our study is important in the context of sustainable development in Oued El Bardi area.   

“In this context, the present study aims to conduct a passive approach of PAHs assessment through bioaccumulation on a plant species, Cupressus sempervirens (a bioaccumulative, perennial, local species resistant to adverse climatic conditions) and one lichen species, Xanthoria parietina (which reacts to smaller doses of pollutants and thus makes it possible to characterize in a practical and safe way the state of an ecosystem) in the industrial zone of Oued El Berdi, Bouira, in Algeria where sustainable development requires a consistent supply of fuel. These two species were selected based on characteristics of bioaccumulation defined by the literature [33,38]. There are easily identifiable and ubiquitous well-known bioaccumulators of air pollutants.”

Reviewer 3 Report

The manuscript presents a study performed to evaluate the concentration of PAHs and PMs accumulated in leaves, barks and thallus used as biomonitors in Oued El Bardi, Algeria. The results of the study are of interest, but I have some comments. First of all, in all the text and Tables, the authors should verify the units of measure and indicate always if they refer to dw. Below, the specific comments.

TITLE: The authors should add Algeria after the name of the monitoring area.

ABSTRACT: The authors should report some numerical result with its p-value. In addition, the authors should report the meaning for Cha,Chb, Chab.

KEY WORDS: The authors should add “biomonitoring” as a key word.

INTRODUCTION SECTION:

- The following sentence needs for a reference: “The monitoring of PAHs is becoming a necessity in environmental researches due to 58 their mutagenic and carcinogenic properties that can seriously threaten human health.”. Cite, for example, the recent review on this issue: https://doi.org/10.1016/j.chemosphere.2022.133948.

- The authors declared that “No investigations were conducted in Africa.”, but right after, they declared that “Lichens constitute another biomonitoring approach. This method was used to detect road traffic pollution [31] as shown by studies conducted in several countries including in Algeria [32].” The authors should clarify what they mean.

- The authors declared that lichens were used to detect road traffic pollution, but lichens were used also for biomonitoring pollution derived from industry, landfills, incinerators, etc. The authors should report this point in their introduction with relative references. Cite, for example: https://doi.org/10.1016/j.envpol.2019.113013, https://doi.org/10.1007/s00244-013-9965-6, https://doi.org/10.1007/s00128-015-1614-5.

MATERIAL AND METHODS SECTION:

- The authors should specify if they determine PM and PAH just in lichen samples. In addition, authors should specify if they determine proline and chlorophyll just in cupressus samples. This point should be better clarify also in the Introduction Section.

- The authors should briefly describe the sites and I suggest to name the sites with the type of pollution source.

- It is not clear what is the method used to determine respectively PM and PAH. The authors should clarify this point.

DISCUSSION AND CONCLUSIONS

- The authors declared that “Findings support previous results having shown the usefulness of lichens and conifers biomonitoring to estimate the local concentrations of PAHs and PMs.”, but they did not report the relevant references.

- Line 330-348: these sentences should be reported (in a more abbreviated form in the Introduction Section, when the authors describe the characteristics of the biomonitors).

- The authors should add the main limitations of their study and, in conclusion wath their findings add to the literature in this field.

FIGURE 1: The authors should report the method used for determining PM and PAH.

TABLE 1: Data reported in Table 1 should be reported in a figure representing a maps with the sites monitored.

TABLE 2: In the legend of the Table 2 is reported the meaning for PM, but in the Table, it is not reported.

TABLE 3: The authors should report the legend for the abbreviations.

Minor editing of English language are required

Author Response

Reviewer 3 Comments

Thank you very much for taking the time to review this manuscript. Please find the detailed responses below and the corresponding revisions.

Comments and Suggestions for Authors

The manuscript presents a study performed to evaluate the concentration of PAHs and PMs accumulated in leaves, barks and thallus used as biomonitors in Oued El Bardi, Algeria. The results of the study are of interest, but I have some comments. First of all, in all the text and Tables, the authors should verify the units of measure and indicate always if they refer to dw. Below, the specific comments.

TITLE: The authors should add Algeria after the name of the monitoring area.

As you suggested, we have added (Algeria) after Bouira (the name of the monitoring area).  

ABSTRACT: The authors should report some numerical result with its p-value.

Thank you for pointing this out. We agree completely that it miss numerical result in our abstract. Accordingly, and as you can see in revised version of our paper, we reported some numerical results with their p-value.

In addition, the authors should report the meaning for Cha,Chb, Chab.

Thank you for pointing this out. We agree and reported the meaning of Cha,Chb, Chab in the revised version of the manuscript.

KEY WORDS: The authors should add “biomonitoring” as a key word.

As suggested, we added “biomonitoring” as a key word just before “bioaccumulation. 

INTRODUCTION SECTION:

- The following sentence needs for a reference: “The monitoring of PAHs is becoming a necessity in environmental researches due to 58 their mutagenic and carcinogenic properties that can seriously threaten human health.”. Cite, for example, the recent review on this issue: https://doi.org/10.1016/j.chemosphere.2022.133948.

Thank you for pointing this out and for your contribution with this valuable reference. As you can see in the revised version of our manuscript, we put this reference in [24].

[24] Abdel-Shafy H.I., Mansour M.S.M., A review on polycyclic aromatic hydrocarbons: Source, environmental impact, effect on human health and remediation, Egypt. J. Petrol. (2015), http://dx.doi.org/10.1016/j.ejpe.2015.03.011

- The authors declared that “No investigations were conducted in Africa.”, but right after, they declared that “Lichens constitute another biomonitoring approach. This method was used to detect road traffic pollution [31] as shown by studies conducted in several countries including in Algeria [32].” The authors should clarify what they mean.

In the sentence before “No investigations were conducted in Africa” we wrote : “So far, assessment of the levels of PAHs through plant biomonitoring have been limited to developed countries [25-29] and some other developing countries as China [30]”.

No investigations were conducted in Africa concerning PAHs (organic pollutants) bioaccumulation.

Studies carried out in Africa about biomonitoring consist on inorganic pollutants. 

- The authors declared that lichens were used to detect road traffic pollution, but lichens were used also for biomonitoring pollution derived from industry, landfills, incinerators, etc. The authors should report this point in their introduction with relative references. Cite, for example: https://doi.org/10.1016/j.envpol.2019.113013 , https://doi.org/10.1007/s00244-013-9965-6 , https://doi.org/10.1007/s00128-015-1614-5.

Thank you for pointing this out and for your contribution with these valuable references. Therefore, we replaced the paragraph by the text below that it looks more complete.

Lichens constitute another biomonitoring approach. This method was used to detect road traffic pollution [32] as shown by studies conducted in several countries including in Algeria [33]. In others studies, this was used the biomonitoring of air pollution derived from industry, landfills, incinerators, etc [34-36].

[34] Vitali, M., Antonucci, A., Owczarek, M., Guidotti, M., Astolfi, M. L., Manigrasso, M., … Protano, C. (2019). Air quality assessment in different environmental scenarios by the determination of typical heavy metals and Persistent Organic Pollutants in native lichen Xanthoria parietina. Environmental Pollution, 254,113013.

doi:10.1016/j.envpol.2019.113013 

[35] Protano, C., Guidotti, M., Owczarek, M. et al. Polycyclic Aromatic Hydrocarbons and Metals in Transplanted Lichen (Pseudovernia furfuracea) at Sites Adjacent to a Solid-waste Landfill in Central Italy. Arch Environ Contam Toxicol 66, 471–481 (2014). https://doi.org/10.1007/s00244-013-9965-6

[36] Protano, C., Owczarek, M., Fantozzi, L. et al. Transplanted Lichen Pseudovernia furfuracea as a Multi-Tracer Monitoring Tool Near a Solid Waste Incinerator in Italy: Assessment of Airborne Incinerator-Related Pollutants. Bull Environ Contam Toxicol 95, 644–653 (2015). https://doi.org/10.1007/s00128-015-1614-5

MATERIAL AND METHODS SECTION:

- The authors should specify if they determine PM and PAH just in lichen samples. In addition, authors should specify if they determine proline and chlorophyll just in cupressus samples. This point should be better clarify also in the Introduction Section.

Thank you for pointing this out. Therefore, we added the paragraph below just at the beginning of Materials and methods section.

The aim of this study was to compare PAH concentrations in Xanthoria parietina with PAH concentrations in Cupressus sepervirens (scales and barks) and PAH concentrations in soil.

In parallel, PM concentrations were compared, as in the case of PAH concentrations, between Xanthoria parietina samples and Cupressus sempervirens (scales and barks separately) samples.

The physiological parameters of chlorophyll and proline were also studied. Thus, for proline concentrations, all lichen samples were compared with conifer scales and conifer barks. Regarding chlorophyll, concentrations in lichen were compared with conifer scales concentrations.    

- The authors should briefly describe the sites and I suggest to name the sites with the type of pollution source.

As recommended, we added the a small description of the sites in the paragraph below: 

The study was mainly developed in the industrialized region of Oued El Berdi. So, five sites SOB1 (Oued El Berdi 1…5) where chosen in there to take samples according to the presence of the studied species (Xanthoria Parietina and Cupressus sempervirens). The description of these sites is detailed in Table 1.

For the caparison, samples of studied species were taken from the control site of Tikejda (TS) and the urban site of Errich (Figure 2).

- It is not clear what is the method used to determine respectively PM and PAH. The authors should clarify this point.

We improved figure 1 and we added the sentence below to clarify this point.

The method used to PM extraction is the classic method which use chloroform to dissolve waxes in vegetal samples.

PM extraction

PAH extraction

DISCUSSION AND CONCLUSIONS

- The authors declared that “Findings support previous results having shown the usefulness of lichens and conifers biomonitoring to estimate the local concentrations of PAHs and PMs.”, but they did not report the relevant references.

Thank you for pointing this out. Therefore, we added the two references below as you can see in the revised version.

[11] Mukhopadhyay, S; Dutta, R; Das, P. A critical review on plant biomonitors for determination of polycyclic aromatic hydrocarbons (PAHs) in air through solvent extraction techniques. Chemosphere 2020, 251.

[29] Terzaghi E., Wild E., Zacchello G., Cerabolini B.E.L., Jones K.C., Di Guardo A., (2013). Forest Filter Effect: Role of leaves in capturing/releasing air particulate matter and its associated PAHs. Atmospheric Environment 74 : 378-384.

- Line 330-348: these sentences should be reported (in a more abbreviated form in the Introduction Section, when the authors describe the characteristics of the biomonitors).

Thank you for pointing this out. Therefore, we summarized this part in the one sentence below and that we inserted in the introduction, where we describe the characteristics of the biomonitors. Just after the sentence, we put the reference [26].

The methods using plants for biomonitoring of air quality may turn out to be successful , as they are economical environmental tool and can supplement the classical physico-chemical methods [26]. 

[26] Badamasi, H. Biomonitoring of Air pollution using plants. J. Environ. Sci. 2017, 2, 27–39.

- The authors should add the main limitations of their study, and in conclusion what their findings add to the literature in this field.

As recommended, we added the limitations of our study in the paragraph below:

We were unable to explain enough the relationships of PM concentrations and PAH concentrations in different sites of our study due to the fact that we did not know exactly the source of each of these pollutants. This needs the chemical characterization the pollutants.

Than we reformulated the advantage of using the biomonitoring method in our study by the following sentence:  

This is a preliminary study that was carried out on the effect of air pollution on vegetation in the industrial region of Oued El Berdi (Bouira) using an economical and environmental method of biomonitoring.

FIGURE 1: The authors should report the method used for determining PM and PAH.

As you suggested here, we substituted “PMs sample preparation technique” by “PM extraction method” and “PAHs sample preparation technique of Soxhlet” by “Soxhlet technique of PAHs extraction”.

TABLE 1: Data reported in Table 1 should be reported in a figure representing a maps with the sites monitored.

TABLE 2: In the legend of the Table 2 is reported the meaning for PM, but in the Table, it is not reported.

We agree with this comment. Therefore, we deleted PM in the legend of table 2 in the revised version. In the first draft of our manuscript table 2 and table 3 were combined into one table.

TABLE 3: The authors should report the legend for the abbreviations.

We agree with this comment. Therefore, we reported the legend for the abbreviations of table 3 in the revised version. In the first draft of our manuscript table 2 and three were combined into one table.

We did some English language rectifactions like “thallus” that we replaced by “thalli” and we write the name of species with forma “italic”. 

Round 2

Reviewer 2 Report

Thank you for the improvement!

Author Response

Thank you very much for taking the time to re-review again this manuscript!

Reviewer 3 Report

The authors accepted almost all of my comments and edited the manuscript accordingly. In my opinion, the manuscript has improved and deserves to be published. I have only two further comments:

- authors should move the sentences added at the beginning of materials and methods to the end of the introduction;

- authors must add "dw" also in the abstract and in all places of the text and tables where necessary.

Minor editing of English language required.

Author Response

Thank you very much for taking the time to re-review again this manuscript. Please find the detailed responses below and the corresponding revisions:

Comment 1.

authors should move the sentences added at the beginning of materials and methods to the end of the introduction;

                     Response to comment 1.

As recommended, we moved the sentence from the beginning of material and methods to the end of the introduction with some rephrasing to be adapted to this position. 

Now, it looks like below:

"The aim of the present study is to assess air pollution in the industrial area of Oued El Berdi, Algeria. It consists of comparing PAH concentrations accumulated in Xanthoria parietina to those accumulated in Cupressus sempervirens (scales and barks) and to that accumulated in soil.

This is also consists to compare PM concentrations between the different compartments (Xanthoria parietina and Cupressus sempervirens (scales and barks separately)) except the soil.

An other aim is to measure the effect of these pollutants on vegetation in dosing chlorophyll and proline content in our lichen and our conifer scales."

Comment 2. 

authors must add "dw" also in the abstract and in all places of the text and tables where necessary.

Response to comment 2. 

Thank you for pointing this out. Accordingly and as you can see in the new version of our manuscript, we added "dw" where necessry.
